# Characterizing two outbreak waves of COVID-19 in Spain using phenomenological epidemic modelling

**Miguel López** 🄳 \*, **Alberto Peinado, Andrés Ortiz**

Department of Telecommunications Engineering, University of Malaga, Malaga, Spain

\* m.lopez@uma.es

**Data Availability Statement:** Data are available from datos.gob.es (https://datos.gob.es/es/catalogo/e05070101-evolucion-de-enfermedad-por-el-coronavirus-covid-19), the World Health Organization (https://www.who.int/emergencies/diseases/novel-coronavirus-2019/situation-

## Abstract

Since the first case reported of SARS-CoV-2 the end of December 2019 in China, the number of cases quickly climbed following an exponential growth trend, demonstrating that a *global pandemic* is possible. As of December 3, 2020, the total number of cases reported are around 65,527,000 contagions worldwide, and 1,524,000 deaths affecting 218 countries and territories. In this scenario, Spain is one of the countries that has suffered in a hard way, the ongoing epidemic caused by the novel coronavirus SARS-CoV-2, namely COVID-19 disease. In this paper, we present the utilization of phenomenological epidemic models to characterize the two first outbreak waves of COVID-19 in Spain. The study is driven using a two-step phenomenological epidemic approach. First, we use a simple generalized growth model to fit the main parameters at the early epidemic phase; later, we apply our previous finding over a logistic growth model to that characterize both waves completely. The results show that even in the absence of accurate data series, it is possible to characterize the curves of case incidence, and construct a short-term forecast of 60 days in the near time horizon, in relation to the expected total duration of the pandemic.

## Introduction

An infectious disease is characterized because it can be transmitted, either directly or indirectly, from one individual to another within a given population. This transmission is materialized in an increase in the number of newly infected individuals per unit time, representing the infectious disease incidence. For example, an infected individual may remain asymptomatic at the early stage of infection, only later developing clinical symptoms, being diagnosed as a disease case. If the number of cases rises above the usual average within a short period of time, a *disease outbreak* occurs, but if the disease spreads quickly to many people, it is an *epidemic*. When the disease persists, and may remain in the population over a long period of time, the disease is said to be *endemic* in the population. Finally, if the disease spreads spatially on a global scale to many countries and continents, a *global pandemic* occurs [1].

At the end of December 2019, a set of pneumonia cases arisen in the city of Wuhan (Hubei province, China). On January 7, 2020, the Chinese authorities notified a new virus from the Coronavirus family as the probably causative agent of the outbreak namely SARS-CoV-2. The

reports), and Worldometer (https://www.
worldometers.info/coronavirus/country/spain/).

**Funding:** This work was partly supported by the
MINECO/FEDER under the PGC2018-098813-B-
C32 project. The funders had no role in study
design, data collection and analysis, decision to
publish, or preparation of the manuscript.

**Competing interests:** The authors have declared
that no competing interests exist.

disease caused by this new virus has been called by international consensus COVID-19, and
the World Health Organization (WHO) recognized it as a global pandemic on 11 March 2020
[2]. As of December 3, 2020, the total number of COVID-19 the reported cases are around
65,527,000 contagions worldwide, and 1,524,000 deaths affecting 218 countries and territories.

Nowadays, the ongoing epidemic caused by the novel coronavirus SARS-CoV-2, represents
the more recent global pandemic arisen in decades; demonstrating that a *global pandemic* is
possible, showing its potential to generate explosive outbreaks not only in cities or regions, but
also in countries or continents following human mobility patterns [3]. While the symptoms
observed in COVID-19 infected people are frequently mild, and quite similar to other com-
mon respiratory infections, it has also exhibited an ability to generate severe disease among
certain groups, including older populations and individuals with underlying health issues such
as cardiovascular disease and diabetes [4]. Nevertheless, the transmission patterns and other
epidemiology characteristics of this novel coronavirus are still being studied.

In this scenario, Spain is one of the countries that has suffered the COVID-19 pandemic in
a hard way. The presence of the virus was first confirmed in Spain on January 31, 2020. Since
then, the number of cases quickly climbed following an exponential growth trend, reaching all
provinces at March 13, 2020. At March 14, 2020, the Spanish Government imposed a confine-
ment or lockdown that was maintained until June 21, 2020 [5, 6]. This first epidemic outbreak
wave, reached its peak at March 20 with 10,651 cases reported and 996 deaths at April 2. The
total cumulative number of cases as of June 21, is of 255,547, and 29,353 deaths [7]. After the
end of the confinement, it was observed a period of stability in the incidence of the disease
accounting for a low number of reported daily cases around the country. However, the num-
ber of new cases starts climbed in July in various cities, presenting again an exponential growth
trend profile, which has remained during the last months. As of December 3, 2020, the total
number of COVID-19 the total cumulative number of cases accounts around 1,693,600 conta-
gions, including 46,000 deaths [5]. This is hence, the so called second wave which is currently
ongoing.

This scenario drive epidemiologist, mathematicians and other scientist to study and fore-
cast the expected behavior of the epidemic outbreak and its dynamics. These studies are based
on classical as well as in phenomenological epidemic models. For example, in [8] the authors
presents an easy way to explain and simplify the mathematics behind some of these compart-
mental epidemiological models. First, they use the most relevant mathematical model Suscep-
tible, Infectious, Removed (SIR) as introduction. Hence, they present other models worth
exploring are the Susceptible, Exposed, Infectious, Removed (SEIR) and the Susceptible,
Unquarantined, Quarantined, Confirmed (SUQC) model. In [9] the authors propose a formu-
lation of COVID-19 based on compartmental models, using partial differential equations
(PDEs). The model is analyzed mathematically, presenting several results concerning its stabil-
ity and sensitivity to different parameters, that may be useful to the understanding of the
COVID-19 behavior and in the interdisciplinary collaboration. In [10] a compartmental math-
ematical model is proposed to address the spread of the COVID-19 for the outbreak that
occurred in Wuhan, China. This paper, focuses on the transmissibility of super-spreaders indi-
viduals. The basic reproduction number threshold is computed and studied respect the local
stability of the disease-free equilibrium in terms of the basic reproduction number. Focusing
on Spain, in [11] the authors present a model for the simulation of COVID-19 based on the
Suspected, Asymptomatic, Infected, Retired (SAIR) compartmental model, by including
mobility of the flow of people between different regions. The numerical experiments show the
importance of data completion and indicate that the model is able to qualitatively simulate the
spread tendencies of small outbreaks. On the side of phenomenological models, in [12] the
authors generate and assess short-term forecasts of the cumulative number of confirmed

reported cases in Hubei province, and for the overall trajectory in China. The forecasts reported in that paper, based on data up until February 9, 2020, largely agree across the three models presented. In [13] it is presented the first study to report the reproduction number of COVID-19 in South Korea, by characterizing the daily local case incidence using a Generalized Growth Model (GGM). In [14] the authors utilize a phenomenological procedure to adjust the number of infected (and removed) individuals in time for 11 countries. The results are obtained by adequately extrapolating a chosen subset of the daily provided data. This phenomenological approach obtains the evolution of the COVID-19 pandemic without using any a priori model based on differential equations. In [15] a study based on Auto-Regressive Integrated Moving Average (ARIMA) is presented. The aim is to predict the epidemiological trend of COVID-19 prevalence of Italy, Spain, and France. The study shows that this kind of models are suitable for predicting the prevalence of COVID-19, and to understand the trends of the outbreak and give an idea of the epidemiological stage of these regions. Finally, in [16] the authors calibrate for the whole of China, 29 provinces in China, and 33 countries and regions four phenomenological models such as the logistic growth model (LGM), the generalized logistic growth model (GGM), the generalized Richards model (GRM) and the generalized growth model (GGM). They use the reported number of infected cases, in order quantitatively document four phases of the outbreak in China with a detailed analysis on the heterogeneous situations across provinces.

In general, all of these studies are intended to assess public health authorities to take appropriate decisions about control measures strategies, in order to mitigate the impact of the disease in the absence of an effective vaccine.

In this paper, we present the utilization of phenomenological epidemic models to characterize the two first outbreak waves of COVID-19 in Spain. The study is driven using a two-step approach to fit the main parameters that characterize both waves. First, a simple growth model is used to identify the intrinsic growth rate and the deceleration parameter for the first and the second wave, during the early stage of the epidemic. Further, these parameters are applied in a second step using a logistic growth model, obtaining a complete characterization of the first wave, and a forecast of the behavior of the second wave. Our main goal is, through simulations and empirical data, to understand and characterize the behavior of the epidemics waves outbreaks of the COVID-19 in Spain. Also, we estimate the possible pandemic cessation dates, and the total size of the epidemics at the end of the second wav by constructing mean-time forecasting.

Hence, this study could be regarded as a valuable tool for characterizing the transmission dynamics process of COVID-19 pandemic along with the impact of control measures, when they were fully implemented and sustained.

The rest of the paper is organized as follows. In Section 2 introduces the basics of phenomenological epidemic models. Section 3 presents the parameter estimation process. Simulation study based on the reported case incidence is presented in Section 4. Section 5 gives the results and discussion; and finally, future works are proposed in Section 6.

## Basics of phenomenological epidemic models

Classical epidemic models are based on mechanistic or physical approaches in order to identify patterns in the observed data, comprising the laws involved in the dynamics of the population or transmission dynamics of a disease. These models such as Kermack and McKendrick [17] or Anderson and May [18] are based on compartments of population in which it is assumed that in the absence of control strategies, the growth dynamics in the first stage of an epidemic is exponential. By contrast, phenomenological epidemic models provide an empirical

approach without a specific basis on the physical laws or mechanisms that give rise to the observed patterns in the data. Most of commonly used phenomenological epidemic approaches are based on population growth models [19].

## The Generalized Growth Model (GGM)

The Generalized Growth Model (GGM) is based on phenomenological population growth dynamics. The model emphasizes in the reproducibility of empirical observations and have been proven to be very useful to model early epidemics in various diseases such as SARS, Ebola, Zika or COVID-19 [12, 13, 20–22]. This model assumes sub-exponential growth dynamic at the beginning of the disease rather than exponential. The GGM relies on two parameters to relax the assumption of exponential growth and is defined by the differential Eq (1) as follows [23]:

$$\frac{dC(t)}{dt} = C'(t) = rC^p(t) \tag{1}$$

Here, $C'(t)$ represents the incidence at time $t$, whereas the solution $C(t)$ is the cumulative number of cases at time $t$. The parameter $r$ is a positive value denoting the intrinsic growth rate $(1/t)$, and the parameter $p$ is considered the *deceleration of growth* factor where $p \in [0, 1]$. If $p = 0$ this differential equation describes a constant incidence curve over time, where the cumulative number of cases grows linearly. By contrast, if $p = 1$ it models exponential growth dynamics, and the solution of the equation is $C(t) = C_0 e^{rt}$, where $C_0$ is the initial number of cases. Choosing intermediate values of $p$ whiting 0 and 1, the equation describes sub-exponential growth behavior. For example, for $p = 1/2$ incidence grows linearly while the cumulative number of cases follows a quadratic polynomial. For $p = 2/3$ incidence grows quadratically while the cumulative number of cases fits a cubic polynomial [20]. This model can also support different epidemic growth patterns depending on the interval in which the value of $p$ moves. This includes linear incidence patterns for $p = 0.5$, concave-up incidence patterns for $0 > p > 0.5$, and concave-down incidence patterns for $0.5 < p < 1$ [20].

## The Generalized Logistic Growth Model (GLGM)

Despite GGM provides an adequate approximation to for the initial phase or *early epidemics* of the transmission dynamics of a disease, this model does not take into account reductions in the disease incidence due to the characteristics of the pathogen, the immunity of the population, or the implementation of medical or social measures. Therefore, consider unlimited growth of contagion is quite unrealistic. To solve this issue, some epidemic models considers logistic growth, in which for a given stable population, it would have a saturation level which representing a numerical *upper bound* for the growth size of the epidemic. This bound is typically called the *carrying capacity K* and in this model, represents the final size of the epidemic. This parameter is crucial to estimate how severe could be the disease, since it represents the final size of the population that became infected. The GLGM is defined by the differential Eq (2) [19, 24]:

$$\frac{dC(t)}{dt} = C'(t) = rC^p(t)\left(1 - \frac{C(t)}{K}\right) \tag{2}$$

where $C(t)$ represents the cumulative number of cases at time $t$, $r$ is the intrinsic growth rate, $p$ is the scaling of growth parameter. As in the GGM $p = 1$ indicates early exponential growth, whereas $p = 0$ represents constant growth, and $0 < p < 1$ accommodates early sub-exponential or polynomial. $K > 0$ is the carrying capacity or final size of the epidemic, which represents

the total number of affected population. This parameter is crucial to generate post-peak forecasts and also, can be linked to an import parameter of classical epidemic models such as the basic reproduction number $R_0$. This parameter represents the transition phase of non–equilibrium process of propagation of a disease. This parameter constitutes a key epidemiological threshold in biological systems since if $R_0 < 1$ the infection dies out while if $R_0 > 1$ it may cause an epidemic disease.

As discussed in [1], there are some standard mathematical compartmental epidemic models. In this context, the Kermack–McKendrick model SIR (*Susceptible-Infected-Recovered*), is a simple model for o study diseases without immunity, that can be used to approximate the basic parameters of a disease such as the reproduction number and the final size. This model is based on a set of differential equations, $dS(t)/dt = -\beta SI$, $dI(t)/dt = \beta SI - \gamma I$, and $dR(t)/dt = \gamma I$. Here, $\beta$ and $\gamma$ are model parameters, and the total population is constant, thus $S(t) + I(t) + R(t) = N$. Hence, possible solutions for the system are $I_0 = 0$; and $I^* > 0$ with $S^* = \gamma/\beta$ or alternatively $1/S^* = \beta/\gamma$. At the beginning of the epidemics $t = 0$, $R(0) = 0$, $I(0) = I_0 << N$, and $S(0) = S_0 = N - I_0$, thus it can be assumed that $S_0 \approx N$ and $1/S_0 = \beta/\gamma$. The relation $S_0\beta/\gamma$ represents the threshold value between the epidemic an endemic situation of the disease, giving the *basic reproductive number $R_0$*.

In fact, assuming that the COVID-19 disease can be modelled as the simplest but useful SIR epidemic model, the basic reproductive number can be calculated as $R_0 = S_0/\rho$, where $\rho = \gamma/\beta = (S_0 - S_\infty)/(ln\ S_0 - ln\ S_\infty)$, $S_0$ is the initial population of susceptible, and $K = S_0 - S_\infty$ is the final size of the epidemic [1]. It should be noted that despite the use of this approach and assumptions possesses some limitations in the calculation of $R_0$, it can provide a great deal as an indicator of severity of the epidemic.

## Parameter estimation

Parameter estimation can be considered as the process of finding parameter values and their confidence intervals that best fit a given model, to the empirical data. In this study, we follow the methodology presented in [12, 25, 26] that has been used to characterize the incidence of various infectious diseases such as Ebola, Smallpox, Measles, or Zika. In essence, this methodology comprises a process that starts with the initial adjustment of the data, then the parameters for each model are fitted based on early epidemic phase. Since these parameter estimations are typically subject to sources of uncertainty, it is constructed the 95% confidence intervals derived from parameter uncertainty.

## Data sources

Since the beginning of the pandemic in Spain, surveillance of cases of this disease is based on the universal notification of all confirmed COVID-19 cases that are identified in each Autonomous Community [27]. The Autonomous Communities (CCAA) make an individualized notification of COVID-19 cases through a web-based computer platform managed by the National Epidemiology Center (CNE). This information comes from the epidemiological case survey that each Autonomous Communities completes upon identification of a COVID-19 case. Despite the fact that some studies have suggested that the number of infections and deaths in Spain may have been underestimated [28], for simulation purposes our study is based on the case incidence reported by the Ministry of Health of Spain, and brought as *csv* file in the official data repository [27]. The time interval comprises 44 weeks (308) days, ranging from January 31, 2020, to December 3, 2020. Fig 1, shows the profile of daily cases reported as blue circles (Fig 1A), as well as the curve of cumulative number of cases blue solid line (Fig 1B). In both pictures, it is easy to identify the two epidemics waves.

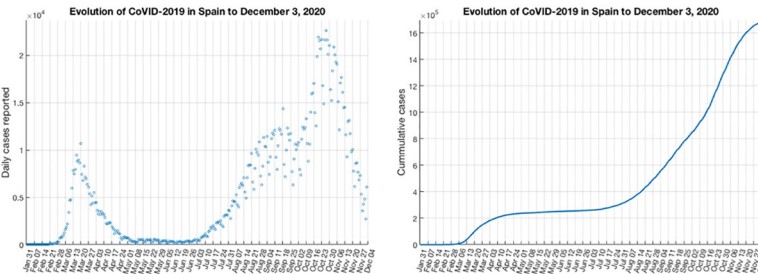

**Fig 1. Cases reported in Spain from January 31, 2020, to December 3, 2020. a.** Profile of daily cases reported. **b.** Curve of cumulative number of cases.

## Parameter fitting

One of the simplest way to estimate the parameters from observed data is using nonlinear least squares fitting. This method tries to obtain the solution that best-fit the curve of the desired model, given a set of data points. For example, applying a non-linear least-squares solver, that uses a *Trust-Region-Reflective* algorithm (or the *Levenberg–Marquardt* algorithm) to provide model coefficients that minimize the sum of squared differences between the model and a set of data [29]. Basically, these methods considers a parameter set $\theta = (\theta_1, \theta_2, \ldots, \theta_k)$, the family of curves $y = f(t,\theta)$ that depend on parameter $\theta$, and a set of data points $(t_1, y_1), (t_2, y_2), \ldots, (t_n, y_n)$. The nonlinear least-squares fitting, tries to find the value of coefficients $\hat{\theta}$ such that the curve $y = f(t, \hat{\theta})$ minimizes the objective function:

$$\hat{\theta} = \min_t \left\| (f(t_i, \theta) - y_{t_i}) \right\|^2 = \min_t \sum_{t=1}^{n} (f(t_i, \theta) - y_{t_i})^2 \tag{3}$$

In order to use a least-squares fitting correctly, it should be defined for the desired parameters $\theta$, the upper and lower bounds, and an initial guess. In the case of GGM $\theta = (r,p)$ and for GLGM $\theta = (r,p,K)$. Also, after the parameters have been estimated, the function provides the residuals, as the difference between the best fit of the model and the time series data as a function of time, in the form $res(t_i) = f(t_i, \hat{\theta}) - y_t$. This is used to assess the quality of the model fit.

## Quantifying parameter confidence intervals

Before the generation of the short-term incidence forecast we still need to quantify the uncertainty of the parameter estimates to construct the 95% confidence intervals, and to identify the potential parameter deviation. One of the methods to quantify parameter uncertainty is the *parametric bootstrapping* [30, 31]. Basically, this method re-estimates the parameters from the previously best-fit model $\hat{\theta}$, by sampling repeatedly multiple observations and generating synthetic datasets. Also, it is considered that the time series follow some distribution error structure (binomial, Poisson, etc.). This method can be implemented in different ways [25, 31]. For this study, we use the parametric bootstrap approach proposed in [25] that has been applied to model various infectious diseases such as Ebola, Smallpox, or Zika [12, 13]. This method assumes that the time series of data follows a Poisson distribution error structure, centered on the mean of the time points of each observation [32, 33]. For a given set of parameter $\hat{\theta}$, a parametric bootstrapping of *m* realizations it is derived a new set of re-estimated parameters $\hat{\theta}_i$, where $i = 1, 2, \ldots, m$. The resulting uncertainty around the new model fit is given by the set of curves $y_1 = f_1(t, \hat{\theta}_1), y_2 = f_2(t, \hat{\theta}_2), \ldots, y_m = f_m(t, \hat{\theta}_m)$.

## Short-term forecasting of incidence curves

The previous calibration of GGM parameters, together the confidence intervals provides a characterization of the early epidemics respect the first and second wave. But we also aim to characterize the profile of the first epidemic wave, and the expected behavior of the second epidemic wave in the near time horizon. In the case of the COVID-19 pandemic, short-term forecast (days or weeks) is useful to develop medical plans or to schedule contention measures in the absence of an effective vaccine. However, it is important to keep in mind that forecasts are often inaccurate as these are mostly based on the current values and uncertainty of the parameters of the system, which are likely to change over time. Moreover, the further out the forecast is made, the more wrong it is expected to be [25]. One method to construct short-term forecast, is to extend the entire uncertainty of the system using the uncertainty associated to the previously computed parameter estimates [25]. In this way, the current state of the system in a given time, is propagated to a time horizon of $T$ time units ahead. Then, the set of curves $y_1 = f_1(t, \hat{\theta}_1), y_2 = f_2(t, \hat{\theta}_2), \ldots, y_m = f_m(t, \hat{\theta}_m)$ are newly computed as $y_1^* = f_1^*(t + T, \hat{\theta}_1)$, $y_2^* = f_2^*(t + T, \hat{\theta}_2), \ldots, y_m^* = f_m^*(t + T, \hat{\theta}_m)$.

## Simulation study

In this simulation, we start estimating the parameters $r$ and $p$ for the first and second wave using the GGM approach. After determining the best fit for $r$ and $p$, now we build the 95% confidence intervals by quantifying the parameter uncertainty. The previous calibration of GGM and the resulting confidence intervals provide a properly adjustment model to data of the early epidemic for the first and second wave. However, GGM only considers unlimited growth of contagion, thus it is not adequate to model an epidemic that has entered or surpassed the peak of contagions.

By contrary, GLGM includes the *upper bound* for the growth size of the epidemic to better characterize the after-peak dynamics of the epidemics, but brings poor adjustment to data in the early stage of the epidemic. Hence, to solve both problems, for the early stage of the epidemic of GLGM we will use the previous parameters and the resulting confidence intervals computed for GGM. Later, assuming that at the time to write this article, the first epidemic wave is finished and that the second wave is ongoing, but approaching to the peak of the maximum case reported, we use the GLGM to better characterize the whole dynamics of the epidemics.

It should be noted that GLGM is quite sensitive to initial definition of guess parameters as well as to the time intervals defined for the early epidemic phase. Therefore, previous to the definition of that values, we have performed some simulations in order to better understand the dynamics of this epidemics.

## Characterization the first wave

**Determining the early epidemic phase.**   To estimate the parameters $r$ and $p$ for early epidemic phase for the first wave using the GGM, based on the data reported it is assumed an early epidemic period of 38 days, (from February 6, 2020 to March 14, 2020), and the initial number of cases reported $C_0 = 11$. The computation of the least-squares fitting is drove setting the guess parameters as $r = 0.5$ and $p = 0.5$, and the bounds of the solution interval as $0 < r < 10$ and $0 < p < 1$ respectively.

Fig 2 shows for the GGM model, the best fit of the early epidemic period for the first wave. The blue circles represent the reported daily case incidence, while the continuous red line corresponds to the best fit of the GGM to the data. The right panel displays the residuals

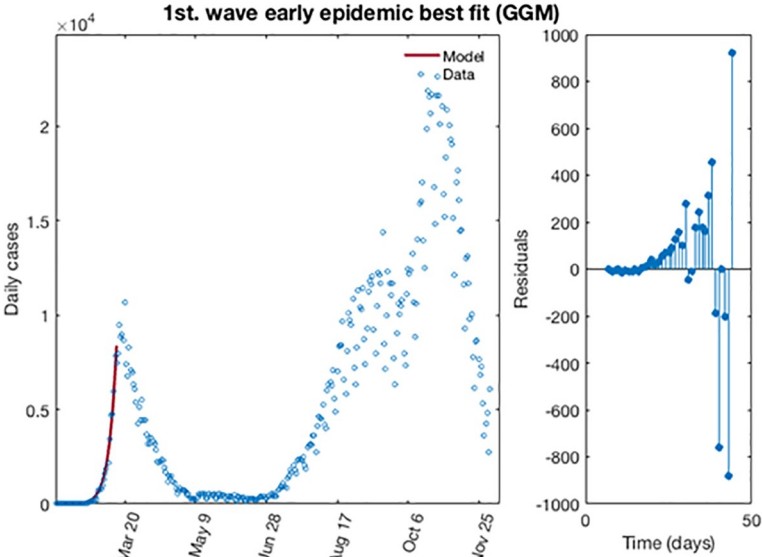

**Fig 2. The best fit for GGM of the early epidemic period for the first wave for COVID-19 in Spain.** Blue circles represent the reported daily case incidence, while continuous red line corresponds to the best fit of the model. The residuals are displayed in the right panel.

$res(t_i) = f(t_i, \hat{\theta}) - y_t$. It should be noted the random pattern of the residuals as a function of time, which suggest that the GGM model has provided a reasonably good fit to the early growth phase of the epidemic.

Now, the first estimation of parameter is then re-fitted using again the GGM. The model is calibrated using $m = 300$ bootstrap realizations, and assuming a Poisson distribution error structure. Also, it has been constructed a short-term forecast of 7 days for the first wave (from, March 14, 2020 to March 20, 2020), by extending the entire uncertainty from the early epidemic period, as described in section 3.4. Fig 3 shows for the GGM model, the resulting curves

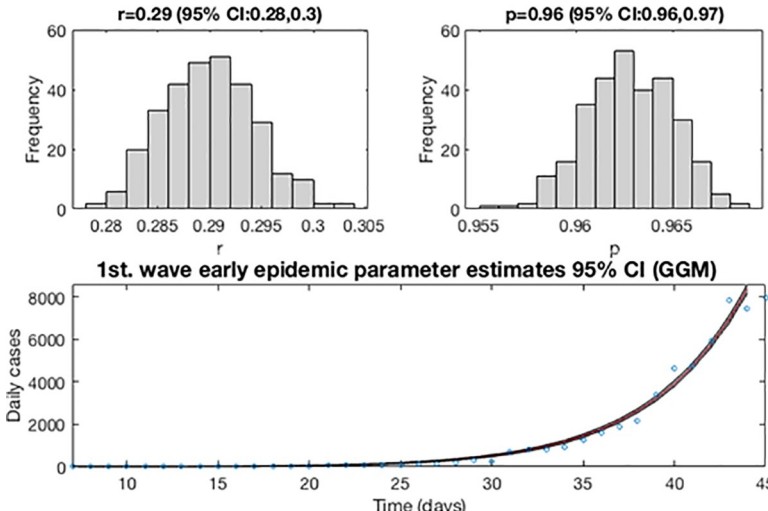

**Fig 3. Resulting curves after re-fitting of parameters of the early epidemic phase.** Blue circles are the cases reported daily; red line corresponds to the best fit of the GGM to the data. Grey lines correspond to the $m$ bootstrapping realizations of the epidemic curves. Black lines correspond to the 95% CI bounds. The empirical distributions for parameter estimations are displayed in the top panel.

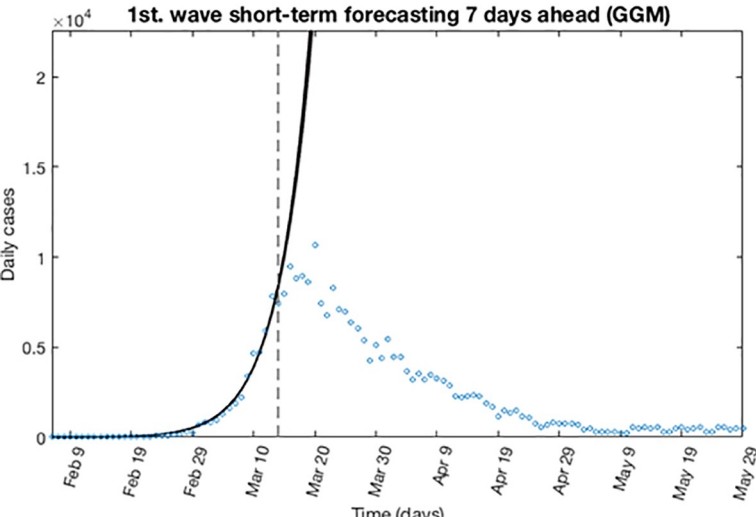

**Fig 4. Short-term forecast.** Forecasted curves for GGM for the first wave using 300 bootstrap realizations. The red line corresponds to the best fit of the forecasted epidemic curve. The black lines correspond to the 95% confidence bounds around the best fit of the forecasted curve, and grey lines correspond to the $m$ bootstrapping realizations of the forecasted epidemic curves. The calibration period is separated from the forecast period with a vertical slashed grey line.

after re-fitting of parameters of the early epidemic phase, and Fig 4 shows a short-term forecast. In the Fig 3, the blue circles are the cases reported daily, while the solid red line corresponds to the best fit of the GGM to the data. The grey lines correspond to the $m$ bootstrapping realizations of the epidemic curves. The black lines correspond to the 95% of confidence interval (CI) bounds around the best fit of the model to the data. The top panels show histograms displaying the empirical distributions for parameter estimations. In the Fig 4, the red line corresponds to the best fit of the forecasted epidemic curve. The black lines correspond to the 95% confidence bounds around the best fit of the forecasted curve, and grey lines correspond to the m bootstrapping realizations of the forecasted epidemic curves. The calibration period is separated from the forecast period with a vertical slashed grey line. All the curves at the right side of this line, correspond to the forecast time horizon. The red line corresponds to the best fit of the forecasted epidemic curve. The black lines correspond to the 95% confidence bounds around the best fit of the forecasted curves, and grey lines correspond to the $m$ bootstrapping realizations of the forecasted epidemic curves.

These Figures show the main limitation of the GGM model, which considers unlimited growth of contagions. This unrealistic assumption, can lead to a wrong extrapolation in the forecasted results, even it is possible to obtain a number of contagions greater than the population under study.

**Determining the complete epidemic dynamics.** In this section, we use the GLGM to better characterize the whole dynamics of the epidemics for the first wave. To start with, now we estimate the best fit of the early epidemic using GLGM approach. For this wave, it is used the previous early period of 38 days, (from February 6, 2020 to March 14, 2020), and the initial number of cases reported $C_0 = 11$. To compute the least-squares fitting, we use the mean of previous parameters estimation from GGM obtained in the previous section. Therefore, the guess for parameters are set as $r = 0.29$ (95% CI: 0.28, 0.30) and $p = 0.96$ (95% CI: 0.96, 0.97), and the bounds was set as $0 < r < 10$ and $0 < p < 1$. Also, we stablish the limits for $K$ at $1 < K < 47.3 \cdot 10^6$, where the upper limit is the population of Spain at January 2020 [27]. Assuming

that *K* is unknown, we estimate an expected final size of the epidemic as of 100,000 cases reported at the end of the first wave.

Fig 5 shows the best fit of the early epidemic period for the first wave. The blue circles represent the reported daily case incidence, while the continuous red line corresponds to the best fit of the GLGM to the data. The right panel displays the residuals $res(t_i) = f(t_i, \hat{\theta}) - y_t$. It should be noted the random pattern of the residuals as a function of time, which suggest that the GGM model has provided a reasonably good fit to the early growth phase of the epidemic.

After determining the best fit, we re-estimate the parameters *r*, *p* and *K* for the GLGM, and build the 95% confidence intervals by quantifying the parameter uncertainty. Fig 6, displays the resulting curves after re-fitting of parameters of the early epidemic phase. The blue circles are the cases reported daily, while the solid red line corresponds to the best fit of the GLGM to the data. The grey lines correspond to the *m* bootstrapping realizations of the epidemic curves. The black lines correspond to the 95% of confidence interval (CI) bounds around the best fit of the model to the data. The top panels show histograms displaying the empirical distributions for parameter estimations.

Based on the previous calculations, now we can determine the epidemic profile for the first wave by extending the entire uncertainty to a concrete period of time as described in section 3.4. Taking into account that the first wave is finished, we can assume to the early epidemic period, a forecast time of 108 days ahead, reaching to June 20, 2020. Figs 7 and 8 show for the GLGM model, the best fit characterization of the first wave using *m* = 300 bootstrap realizations. In Fig 7, the blue circles are the daily cases reported. The solid grey lines correspond to the *m* bootstrapping realizations of the epidemic curves. The solid red line corresponds to the mean confidence or best fit of the GLGM respect the data; while solid black lines correspond to the upper and lower bounds of 95% confidence around the best fit of the model to the early epidemic. Also, in the Fig 8 it has been included, the curve of cumulative number of cases represented in the solid blue line; the red line corresponds to the best fit of the GLGM to the data. Grey lines correspond to the *m* bootstrapping realizations of the epidemic curves. Black lines correspond to the 95% CI bounds.

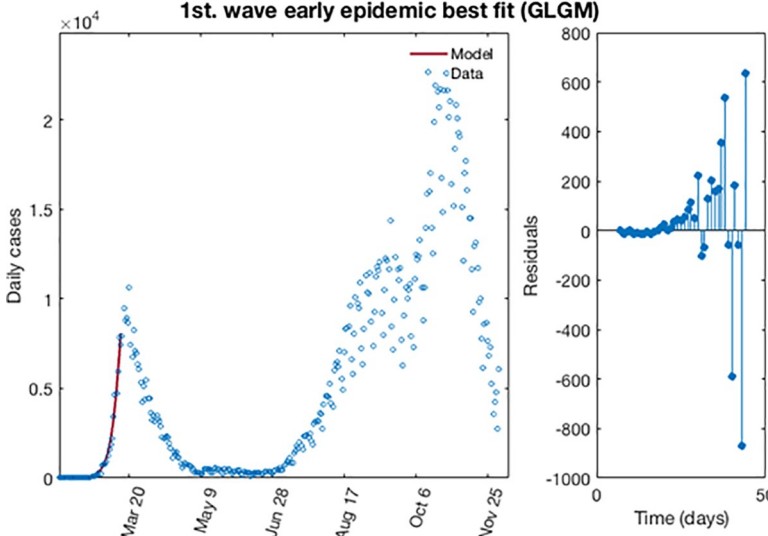

**Fig 5. The best fit for GLGM of the early epidemic period for the first wave for COVID-19 in Spain.** Blue circles represent the reported daily case incidence, while continuous red line corresponds to the best fit of the model. The residuals are displayed in the right panel.

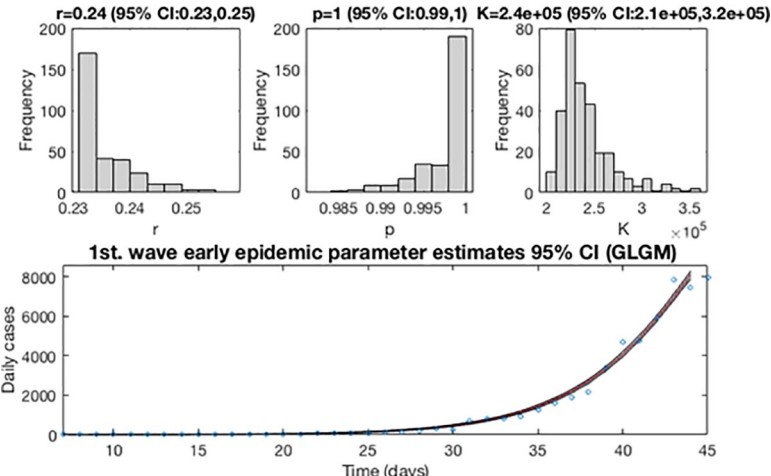

**Fig 6. Resulting curves after re-fitting of parameters of the early epidemic phase.** Blue circles are the cases reported daily; red line corresponds to the best fit of the GLGM to the data. Grey lines correspond to the $m$ bootstrapping realizations of the epidemic curves. Black lines correspond to the 95% CI bounds. The empirical distributions for parameter estimations are displayed in the top panel.

## Characterization of the second wave

**Determining the early epidemic phase.** Based on the data reported, to determine the early epidemic phase for the second wave it is assumed an early epidemic period of 45 days (from, July 30, 2020, to August 14, 2020), and the initial number of cases reported $C_0 = 540$. Also, the computation of the least-squares fitting was drove setting the guess parameters as $r = 0.5$ and $p = 0.5$, and the bounds of the solution interval as $0 < r < 10$ and $0 < p < 1$ respectively. The first estimation of parameter is then re-fitted using again the GGM. The model is calibrated using $m = 300$ bootstrap realizations, and assuming a Poisson distribution error structure.

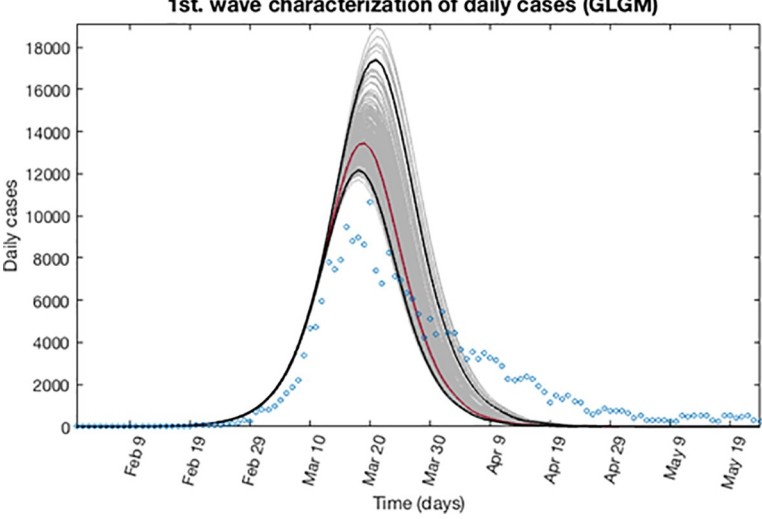

**Fig 7. Characterization of the daily epidemic profile for the first wave using GLGM.** The blue circles are the cases reported daily; red line corresponds to the best fit of the GLGM to the data. Grey lines correspond to the $m$ bootstrapping realizations of the epidemic curves. Black lines correspond to the 95% CI bounds.

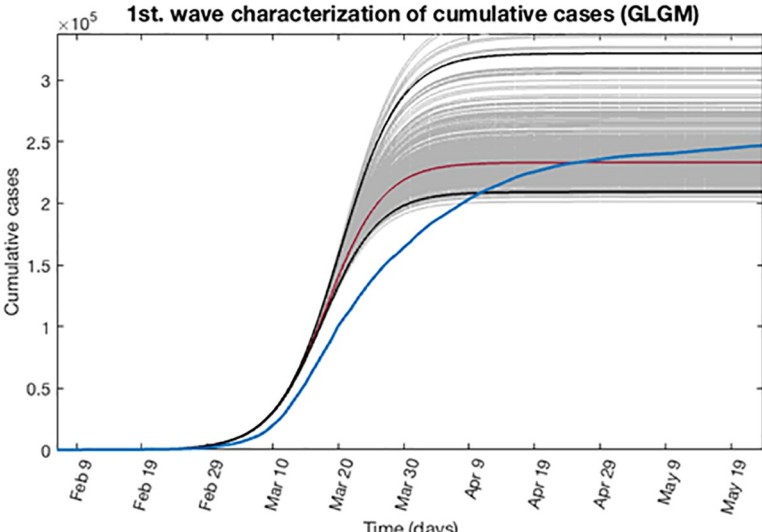

**Fig 8. Characterization of the cumulative epidemic profile for the first wave using GLGM.** The solid blue line is the cumulative number of cases; red line corresponds to the best fit of the GLGM to the data. Grey lines correspond to the $m$ bootstrapping realizations of the epidemic curves. Black lines correspond to the 95% CI bounds.

Fig 9 shows for the GGM model, the best fit of the early epidemic period for the first wave. The blue circles represent the reported daily case incidence, while the continuous red line corresponds to the best fit of the GGM to the data. The right panel displays the residuals $res(t_i) = f(t_i, \hat{\theta}) - y_t$. It should be noted the random pattern of the residuals as a function of time, which suggest that the GGM model has provided a reasonably good fit to the early growth phase of the epidemic.

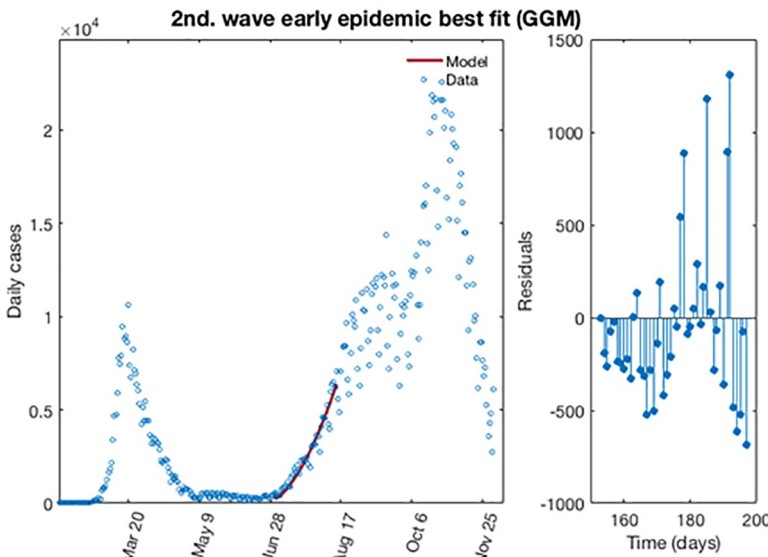

**Fig 9. The best fit for GGM of the early epidemic period for the second wave for COVID-19 in Spain.** Blue circles represent the reported daily case incidence, while continuous red line corresponds to the best fit of the model. The residuals are displayed in the right panel.

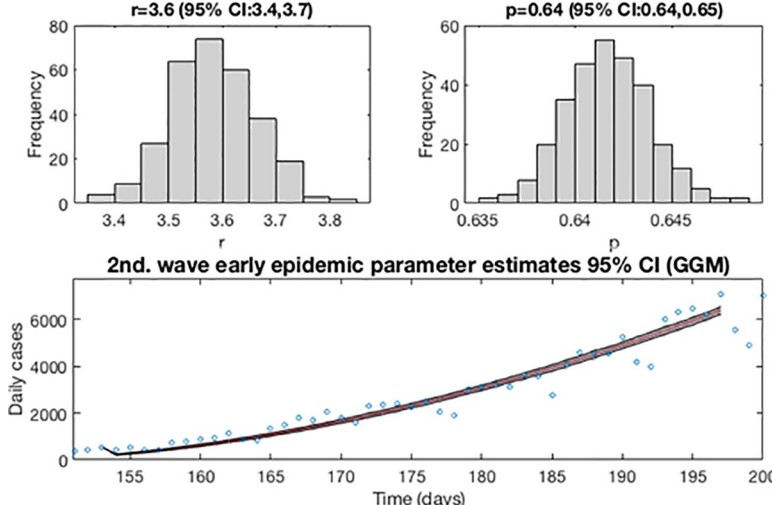

**Fig 10. Resulting curves after re-fitting of parameters of the early epidemic phase.** Blue circles are the cases reported daily; red line corresponds to the best fit of the GLGM to the data. Grey lines correspond to the $m$ bootstrapping realizations of the epidemic curves. Black lines correspond to the 95% CI bounds. The empirical distributions for parameter estimations are displayed in the top panel.

Now, as in the previous wave, the first estimation of parameter is then re-fitted using the GLGM. The model is calibrated using $m = 300$ bootstrap realizations, and assuming a Poisson distribution error structure. Also, it has been constructed a short-term forecast of 21 days (from August 14, 2020, to September 3, 2020), by extending the entire uncertainty from the early epidemic period, as described in section 3.4.

Fig 10 shows for the GLGM model, the resulting curves after re-fitting of parameters of the early epidemic phase and Fig 11 shows the short-term forecast. In the Fig 10, the blue circles are the cases reported daily, while the solid red line corresponds to the best fit of the GLGM to the data. The grey lines correspond to the $m$ bootstrapping realizations of the epidemic curves. The black lines correspond to the 95% of confidence interval (CI) bounds around the best fit of the model to the data. The top panels show histograms displaying the empirical distributions for parameter estimations. In the Fig 11, The red line corresponds to the best fit of the forecasted epidemic curve. The black lines correspond to the 95% confidence bounds around the best fit of the forecasted curve, and grey lines correspond to the m bootstrapping realizations of the forecasted epidemic curves. The calibration period is separated from the forecast period with a vertical slashed grey line. All the curves at the right side of this line, correspond to the forecast time horizon. The red line corresponds to the best fit of the forecasted epidemic curve. The black lines correspond to the 95% confidence bounds around the best fit of the forecasted curves, and grey lines correspond to the $m$ bootstrapping realizations of the forecasted epidemic curves.

**Forecasting the expected epidemic dynamics.** To start with characterization and forecasting of the second wave, now we estimate the best fit of the early epidemic using GLGM approach. As mentioned before, the GLGM is quite sensitive to initial definition of guess parameters as well as to the time intervals defined for the early epidemic phase. To solve this issue, we have performed previous simulations in order to better understand the dynamics of this epidemics, and for the definition of guess parameters. For example, if it is assumed the same early epidemic period as used in the previous section for GGM, the model presents an important decrease of cases in the early epidemic phase. This is due to the fact that the number

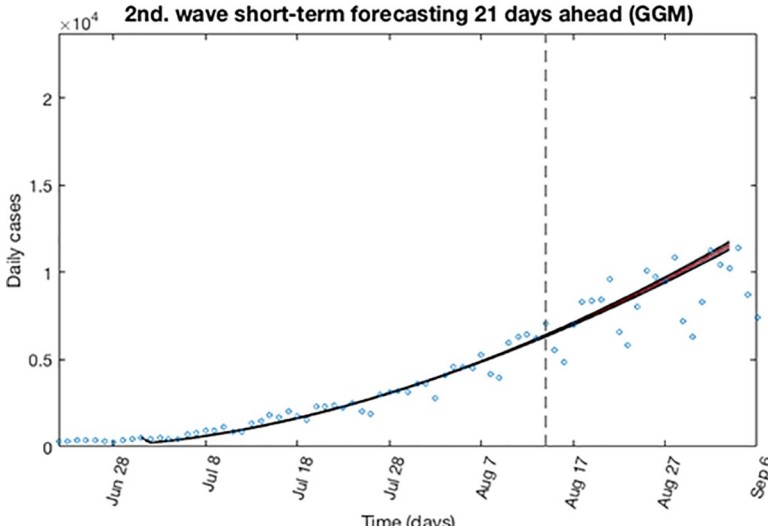

**Fig 11. Short-term forecast curves for GLGM for the second wave using 300 bootstrap realizations.** The red line corresponds to the best fit of the forecasted epidemic curve. The black lines correspond to the 95% confidence bounds around the best fit of the forecasted curve, and grey lines correspond to the $m$ bootstrapping realizations of the forecasted epidemic curves. The calibration period is separated from the forecast period with a vertical slashed grey line.

of daily cases reported during this stage of the epidemics presents high over dispersion, and GLGM needs a minimal correlation between data to be adjusted properly. By contrary, increasing the early epidemic period, the model seems to growth exponentially reaching an expected number of cases higher than the population to study. This is probably due to the previously mentioned over dispersion of data and the underreported cases. In this model, it is assumed an early epidemic period of 55 days (from, July 8, 2020, to September 2, 2020) and the initial number of cases reported $C_0 = 898$. Hence, we use the mean of previous parameters estimation from GGM, setting the guess for parameters $r = 3.6$ (95% CI: 3.40, 3.70) and $p = 0.64$ (95% CI: 0.64, 0.65), and the bounds was set as $0 < r < 10$ and $0 < p < 1$. The limits for $K$ are stablished as $2.4 \cdot 10^5 < K < 47.3 \cdot 10^6$, where the lower limit is the mean value of the final size of the epidemic obtained from the calibration of the early epidemic of the first wave using GLGM. Since the second wave is still ongoing, $K$ is unknown its guess value is stablished as an increase of 50% of the final size of the epidemic obtained from the calibration of the early epidemic of the first wave using GLGM.

Fig 12 shows the best fit of the early epidemic period for the second wave. The blue circles represent the reported daily case incidence, while the continuous red line corresponds to the best fit of the GLGM to the data. The right panel displays the residuals $res(t_i) = f(t_i, \hat{\theta}) - y_t$. It should be noted the random pattern of the residuals as a function of time, which suggest that the GGM model has provided a reasonably good fit to the early growth phase of the epidemic.

Now, after determining the best fit for both waves, we re-estimate the parameters $r$, $p$ and $K$ for the GLGM, and build the 95% confidence intervals by quantifying the parameter uncertainty. Fig 13 displays the re-fitting of parameters of the early epidemic phase for the first and the second wave using the GLGM. The model has been calibrated using $m = 300$ bootstrap realizations, and assuming a Poisson distribution error structure. The blue circles are the cases reported daily, while the solid red line corresponds to the best fit of the GLGM to the data. The grey lines correspond to the $m$ bootstrapping realizations of the epidemic curves. The black lines correspond to the 95% of confidence interval (CI) bounds around the best fit of the

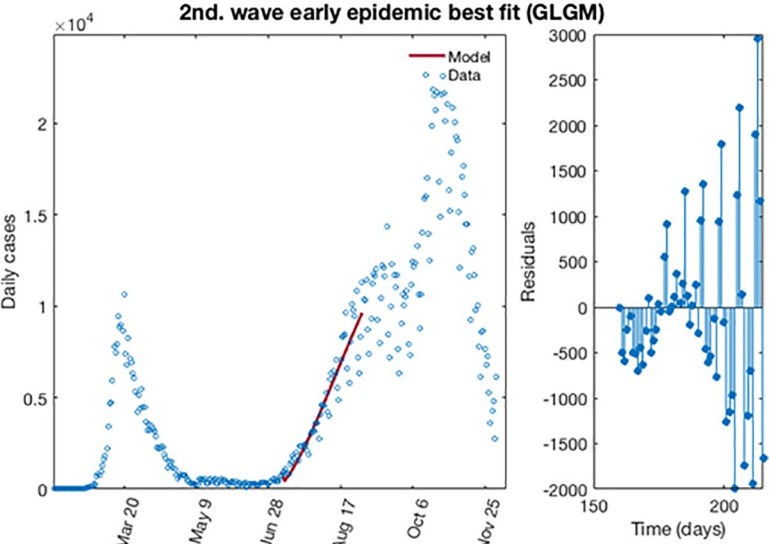

**Fig 12. The best fit for GLGM of the early epidemic period for the second wave for COVID-19 in Spain.** Blue circles represent the reported daily case incidence, while continuous red line corresponds to the best fit of the model. The residuals are displayed in the right panel.

model to the data. The top panels show histograms displaying the empirical distributions for parameter estimations.

Based on the previous calculations, now we can address an expected behavior or forecast for the second wave. Here we determine the epidemic profile by extending the entire uncertainty to a concrete period of time, adding to the early epidemic period, a forecast of 151 days ahead as described in section 3.4. In this case, the forecast starts at September 2, 2020 and finished at January 29, 2021. Figs 14 and 15 show for the GLGM model, the best fit characterization of the first wave using $m = 300$ bootstrap realizations. In Fig 14, the blue circles are the daily cases reported. The solid grey lines correspond to the $m$ bootstrapping realizations of the

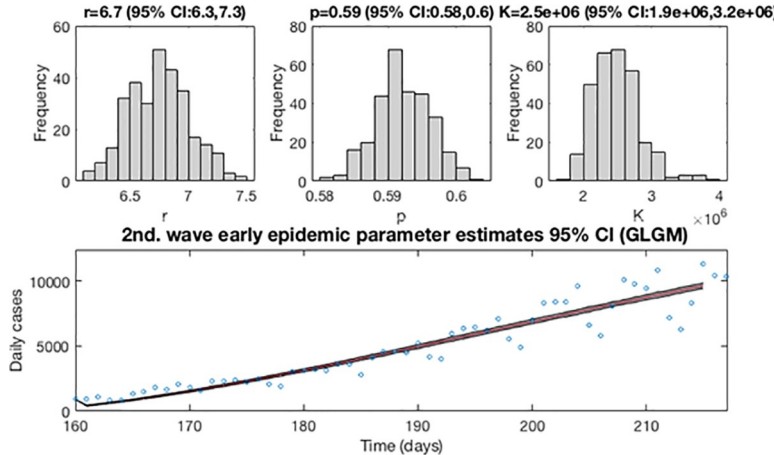

**Fig 13. Resulting curves after re-fitting of parameters of the early epidemic phase.** Blue circles are the cases reported daily; red line corresponds to the best fit of the GLGM to the data. Grey lines correspond to the $m$ bootstrapping realizations of the epidemic curves. Black lines correspond to the 95% CI bounds. The empirical distributions for parameter estimations are displayed in the top panel.

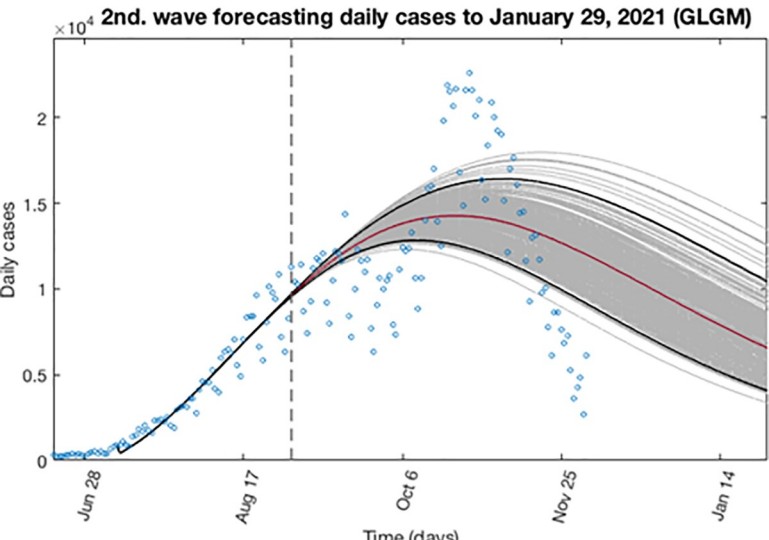

**Fig 14. Characterization of the expected daily epidemic profile for the second wave using GLGM.** The blue circles are the cases reported daily; red line corresponds to the best fit of the GLGM to the data. Grey lines correspond to the *m* bootstrapping realizations of the epidemic curves. Black lines correspond to the 95% CI bounds.

epidemic curves. The solid red line corresponds to the mean confidence or best fit of the GLGM respect the data; while solid black lines correspond to the upper and lower bounds of 95% confidence around the best fit of the model to the early epidemic. Also, it has been included in this figure, the curve of cumulative number of cases represented in Fig 15 as the solid blue line; the red line corresponds to the best fit of the GLGM to the data. Grey lines correspond to the m bootstrapping realizations of the epidemic curves. Black lines correspond to the 95% CI bounds. In both figures, it has been included a vertical slashed gray line to separate the calibration of the early epidemic period, to the forecasted period.

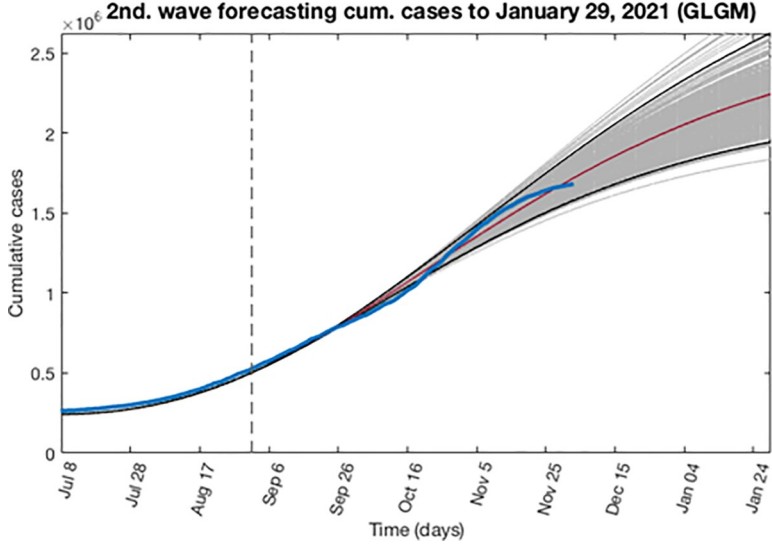

**Fig 15. Characterization of the expected cumulative epidemic profile for the second wave using GLGM.** The solid blue line is the cumulative number of cases; red line corresponds to the best fit of the GLGM to the data. Grey lines correspond to the *m* bootstrapping realizations of the epidemic curves. Black lines correspond to the 95% CI bounds. The vertical slashed gray line to separate the calibration of the early epidemic period, to the forecasted period.

## Results and discussion

Simulation results and the main parameters that characterize both waves are shown in Table 1, bringing important information about the behavior of the epidemics over time. In general, the limitations presented in these models make them only applicable in some stages of the disease. For example, when the epidemic follows an exponential growth at an early stage, the use of the GGM is more suitable for this initial regime. After that, the growth rate decays as susceptible population decrease or if some countermeasures are applied to reduce the transmission of the virus. In that case the GLGM is a better option to characterize the after-peak stage of an epidemic.

For the first wave, the GGM model indicate that its early epidemic dynamics follows an exponential growth dynamics, with values of $p \approx 1$, and the cumulative number of cases can be obtained by the equation $C(t) = C_0 e^{rt}$. In the case of GLGM, the model estimates a final size of the epidemic $K = 2.40 \cdot 10^5$ (95% CI: $2.1 \cdot 10^5$, $3.2 \cdot 10^5$), whereas the data reported is $2.58 \cdot 10^5$ cases. This is displayed in Fig 8. Also, the model is able to detect the peak of the epidemic for that first wave accurately.

In the case of the second wave, we have first estimate the epidemic profile to the last data available at the time to write this article. The GGM and GLGM models indicate that the epidemics curve follows a concave-down incidence pattern ($0.5 < p < 1$). In GGM, it has been obtained values of $p = 0.64$ (95% CI: 0.64, 0.65), which are near to 2/3, predicting that the incidence will grow quadratically, while the cumulative number of cases will follow a cubic polynomial pattern. Likewise, the GLGM model also indicates similar behavior for the epidemic growth, obtaining a value of $p = 0.59$ (95% CI: 0.58, 0.60). In this case, since the second wave is ongoing, the final size of the epidemic is forecasted. Hence, we have extended the entire uncertainty to January 29, 2021, obtaining $K = 2.48 \cdot 10^6$ (95% CI: $1.93 \cdot 10^6$, $3.24 \cdot 10^6$), when the current data indicate that as for December 3, 2020, the cumulative number of cases reported is 1,693,591. The profile of the cumulative number of cases, reported and forecasted, are shown in Fig 14.

**Table 1. Main parameters of epidemic curves.**

| Model | Parameter | 1st. Epidemic Wave (from January 31, 2020 to June 20, 2020) | 2nd. Epidemic Wave (from July 8, 2020 to January 29, 2021) |
|---|---|---|---|
| GGM | *Early period* | 38 days | 45 days |
|  | $C_0$ | 11 | 540 |
|  | $r_{guess}$ | 0.5 | 0.5 |
|  | $p_{guess}$ | 0.5 | 0.5 |
|  | $r_{estimated}$ | 0.29 (95% CI: 0.28, 0.30) | 3.6 (95% CI: 3.40, 3.70) |
|  | $p_{estimated}$ | 0.96 (95% CI: 0.96, 0.97) | 0.64 (95% CI: 0.64, 0.65) |
| GLGM | *Early period* | 38 days | 55 days |
|  | $C_0$ | 11 | 898 |
|  | $r_{guess}$ | 0.29 | 3.6 |
|  | $p_{guess}$ | 0.96 | 0.64 |
|  | $K_{guess}$ | 100,000 cases | 360,000 cases |
|  | $r_{estimated}$ | 0.24 (95% CI: 0.23, 0.25) | 6.8 (95% CI: 6.30, 7.30) |
|  | $p_{estimated}$ | 1.0 (95% CI: 0.99, 1.0) | 0.59 (95% CI: 0.58, 0.60) |
|  | $K_{estimated}$ | $2.40 \cdot 10^5$ (95% CI: $2.1 \cdot 10^5$, $3.2 \cdot 10^5$) | $2.48 \cdot 10^6$ (95% CI: $1.93 \cdot 10^6$, $3.24 \cdot 10^6$) |
| Reported | *Cumulative Cases* | $2.58 \cdot 10^5$ | $1.69 \cdot 10^6$ (as for December 3, 2020) |
| SIR | $R_0 = S_0/\rho$ | 1.0025 | 1.0274 (estimated at January 29, 2021) |

Main parameters that characterize the epidemic curves of the COVID-19 in Spain, from January 31, 2020 to a forecasted period to January 29, 2021.

Overall, our analysis of empirical COVID-19 disease data in Spain, using GLGM model has revealed that, in the absence of an effective vaccine in the near months, the epidemic profile of the second wave can become in the third epidemic wave or, even more, it could remain in the population for long time, probably, to the end of summer 2021. This should be take into account in order to implement strict contention measures.

## Conclusions and future work

In this paper, we have described the utilization of two phenomenological models to understand and characterize shifts in the epidemic growth patterns of the epidemics waves outbreaks of the COVID-19 in Spain. We have illustrated the proposed approach through simulations using data gathering of 44 weeks, and coupling the Generalized Growth Model (GGM) and the Generalized Logistic Growth Model (GLGM). Our findings indicate that even if the information about the disease is could be limited or scattered, using an adequate combination of phenomenological models, it is possible to depict epidemic growth patterns over time and construct short terms forecasts. It should be noted that these models can have some limitations at the time to brig forecasts, due to the use of extrapolation of data and could offer wrong estimations at the long-term forecasts. Despite that limitations, these models are still useful to represents important aspects of a disease, and to understand their behavior. For example, the model shows that in the absence of an effective vaccine, the second wave was likely inevitable and arrived just a few days after the end of the confinement. Likewise, the model shows that the delay in the implementation of strict control interventions at the beginning of this second wave, and the maintenance of these measures, may have caused an excess in the exponential growth of cases disease. This is reflected in the increase of the value of intrinsic growth rate $r$ and, the basic reproductive number $R_0$ of the second wave, respect the first one. Hence, at the current state of the epidemic, it has been demonstrated that the strategy of adaptive or partial confinement may not be effective at all as a control measure, and even that a third wave is possible.

On the other side, underreporting is a common issue in the study of diseases, and in the case of COVID-19 this is a main concern due to a high number of asymptomatic cases. In this context, the basic reproductive number $R_0$ obtained in this study seems to be quite unrealistic, indicating that probably not all infected has been properly diagnosed and accounted.

Finally, the parameter estimations about the COVID-19 obtained in the article are based on phenomenological models, that accounts for the infected individuals. Despite the fact that this is an important start point to give predictions about the growth dynamics of the disease, these models do not show some other aspects of the disease parameters such as the infectious period, incubation period, proportion of asymptomatic, initial amount of recovered people, and so on. Therefore, in future works, our goal is to provide a better approximation of the up to date epidemic curves of the COVID-19, based on phenomenological models together the use of compartmental ones. For this purpose, it is crucial to dispose of accurate and up to date sets of data series of diagnosed, undiagnosed, recovered and reported cases.

## Acknowledgments

Research Group BioSip TIC-251 (Biomedical Signal Processing, Computational Intelligence and Communication Security).

## Author Contributions

**Supervision:** Alberto Peinado, Andrés Ortiz.

**Writing – original draft:** Miguel López.

**Writing – review & editing:** Miguel López.

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
