## [Decision Letter · Decision Letter 0]

7 Apr 2021

PONE-D-20-38562

Characterizing Two Outbreak Waves of COVID-19 in Spain Using Phenomenological Epidemic Modelling

PLOS ONE

Dear Dr. Lopez,

Thank you for submitting your manuscript to PLOS ONE. After careful consideration, we feel that it has merit but does not fully meet PLOS ONE’s publication criteria as it currently stands. Therefore, we invite you to submit a revised version of the manuscript that addresses the points raised during the review process.

Specifically, the reviewer found the work interesting but there are a number of issues that must be addressed before the manuscript can be further considered..

We look forward to receiving your revised manuscript.

Kind regards,

Oscar Millet

Academic Editor

PLOS ONE

Journal Requirements:

2. Please note that in order to use the direct billing option the corresponding author must be affiliated with the chosen institute. Please either amend your manuscript to change the affiliation or corresponding author, or email us at plosone@plos.org with a request to remove this option.

3. Please ensure that you refer to Figure 5 in your text as, if accepted, production will need this reference to link the reader to the figure.

Reviewers' comments:

Reviewer's Responses to Questions

**Comments to the Author**

1. Is the manuscript technically sound, and do the data support the conclusions?

Reviewer #1: Partly

2. Has the statistical analysis been performed appropriately and rigorously? 

Reviewer #1: No

3. Have the authors made all data underlying the findings in their manuscript fully available?

Reviewer #1: Yes

4. Is the manuscript presented in an intelligible fashion and written in standard English?

Reviewer #1: Yes

5. Review Comments to the Author

Reviewer #1: In general, the topic of the article is important. Several aspects need to be improved to avoid wrong conclusions and public health messages. It is important to remark that media, some researchers and people in general are dissatisfied with the forecasting made by several studies since those have been wrong. The idea is to improve the mathematical process and the validation.

These are the main general aspects that I have found that need to be fixed. The authors need to emphasize that the models presented here are only valid for a very short term, and that they do not take into account changes in social behavior and NPI interventions. Second, since it is a characterization, it is important to see if the parameters are identifiable in a unique way. The algorithm used in this article is a local one, so it would be necessary to use a global optimizer or justify the locality. In addition, the authors need to show the bootstrap correlations plots in the parameter space to address identifiability. Finally, in several parts of the article the authors mention that the model can be used for forecasting and talk about final epidemic size. It is important that the authors mention that the model is valid for a short period and under the current COVID-19 pandemic where there are many changes in social behavior, the models cannot forecast the final epidemic size. In the last table there is an estimation of R0 using final epidemic size, this needs to be modified and explained.

Here particular aspects to improve the paper taking into account previous general comments:

There are few references and the introduction can be improved taking into account previous works.

Line 20. How long? Relative to what?

Line 73. Change the word current. If people look, the current situation in Spain would disagree with the result presented here and specially with the 2nd wave.

Line 87 fix the name

Line 141 add reference and assumptions for this equation. This formula includes final epidemic size, which can't be obtained with the proposed models. Please explain this and the limitations.

Line 213. Is a good comment.

Section 4.1. Explain why and how you chose the specifics early epidemic period? Notice that this would change the values of the estimated parameters.

Section 4.. For the first and second waves, the periods are different. Show in table Co for each wave.

The bootstrapping plots (parameter space) can show correlations between the parameters and asses if the a parameters are identifiable. If this is not true, then there are infinite sets of parameter values such the fit is equally good.

Fig 6. The bootstrapping plots (parameter space) can show correlations between the parameters and asses if the parameters are identifiable. If this is not true, then there are infinite sets of parameter values such the fit is equally good. In this case, it seems that parameters are correlated, please show bootstrap plots in the parameter spaces.

Divide the section in 1st wave and 2nd wave to see the differences more clearly.

For each wave summarize initial condition, parameters, period, results in a Table.

Line 377. I would not use final epidemic size here since as the authors mention this method is only valid for the early period. Moreover, as the authors mention this model does not take into account variation on social behaviors and NPI.

Line 406. This paragraph needs to be changed carefully. The models presented here can't do long term forecast. This has been already mentioned by several studies and the media and people in general are blaming the models and forecasting. Just looking the current data of Spain it can be seen that extrapolation is wrong. However, still models are useful for other aspects. Please emphasize limitations of the study, and focus on what can be done with these limited models.

Lines 415. A deeper discussion would be useful regarding the unreported cases and the effect on the Ro and compare with other studies, Underreporting is a common issue an should be discussed. Authors should add some discussion of the asymptomatic cases and the findings n this study. Finally, discuss about the estimations given in the article and how this is affected by the infectious period, incubation period, proportion of asymptomatic, initial amount of recovered people

6. PLOS authors have the option to publish the peer review history of their article (what does this mean?). If published, this will include your full peer review and any attached files.

Reviewer #1: No

---

## [Author Response · Author response to Decision Letter 0]

14 May 2021

Dear Academic Editor of PLOS ONE,

First of all, I like to thank the reviewers for the received comments in order to improve the paper. I have included the corrections according to the reviewer’s comments as shown below, and I have tried to do my best. 

In this revision note, reviewer’s comments are written in orange, and my responses are written in green. I also provide a marked-up copy of my manuscript that highlights the changes made to the original version

We hope that now the paper meets with the requirements to be published in Computer Networks Journal.

Yours sincerely,

Miguel Lopez

---

## [Editor Report · Decision Letter 1]

27 May 2021

Characterizing Two Outbreak Waves of COVID-19 in Spain Using Phenomenological Epidemic Modelling

PONE-D-20-38562R1

Dear Dr. Lopez,

We’re pleased to inform you that your manuscript has been judged scientifically suitable for publication and will be formally accepted for publication once it meets all outstanding technical requirements.

Kind regards,

Oscar Millet

Academic Editor

PLOS ONE
---

## [Editor Report · Acceptance letter]

16 Jun 2021

PONE-D-20-38562R1 

Characterizing two outbreak waves of COVID-19 in Spain using phenomenological epidemic modelling 

Dear Dr. López:

I'm pleased to inform you that your manuscript has been deemed suitable for publication in PLOS ONE. Congratulations! Your manuscript is now with our production department. 

Kind regards, 

on behalf of

Dr. Oscar Millet 

Academic Editor

PLOS ONE